# Etiology of Mastitis and Antimicrobial Resistance in Dairy Cattle Farms in the Western Part of Romania

**DOI:** 10.3390/antibiotics11010057

**Published:** 2022-01-03

**Authors:** Corina Pascu, Viorel Herman, Ionica Iancu, Luminita Costinar

**Affiliations:** Faculty of Veterinary Medicine, Banat’s University of Agricultural Sciences and Veterinary Medicine “King Michael I of Romania” from Timisoara, Calea Aradului 119, 300645 Timisoara, Romania; viorelherman@usab-tm.ro (V.H.); ionica.iancu@usab-tm.ro (I.I.)

**Keywords:** mastitis, etiology, multidrug resistance, Romania

## Abstract

The present study aimed to determine the bacteria isolated from bovine mastitis and their antimicrobial resistance in the western part of Romania. Clinical mastitis was diagnosed based on local inflammation in the udder, changes in milk, and when present, generalized symptoms. Subclinical mastitis was assessed using a rapid test—the California Mastitis Test. The identification of bacterial strains was performed based on biochemical profiles using API system tests (API 20 E, API Staph, API 20 Strep, API Coryne, API 20 NE (bioMerieux, Marcy l’Etoile, France), and MALDI-TOF mass spectrometry (MS). The prevalent isolated bacteria were *Staphylococcus* spp. (50/116; 43.19%), followed by *Streptococcus* spp. (26/116; 22.41%), *E. coli* (16/116; 13.79%), *Corynebacterium* spp. (9/116; 7.75%), *Enterococcus* spp. (10/116; 8.62%), and *Enterobacter* spp. (5/116; 4.31%). Phenotype antimicrobial resistance profiling was performed used the disc diffusion method. Generally, Gram-positive bacteria showed low susceptibility to most of the antimicrobials tested, except cephalothin. Susceptibilities to penicillins and quinolones were fairly high in Gram-negative bacteria, whereas resistance was observed to macrolides, aminoglycosides, and tetracyclines. The highest number of isolates were multidrug resistant (MDR), the resistance pathotypes identified including the most frequently antimicrobials used in cow mastitis treatment in Romania.

## 1. Introduction

Bovine mastitis is the foremost endemic infectious disease of dairy cattle worldwide, as well as in our country. Mastitis is responsible for major economic losses to dairy producers and the milk processing industry, resulting in reduced milk production, alteration in milk composition, discarded milk, increased replacement costs, treatment costs, and veterinary services [1]. Apart from the substantial economic losses associated with the disease, mastitis has serious zoonotic potential and has been associated with the increasing development and the rapid emergence of multidrug resistant strains globally [2,3,4,5]. The welfare implications of mastitis are severe and were highlighted in reports in different countries.

Mastitis, the inflammation of the mammary gland, usually a consequence of the adhesion, invasion, and colonization of the mammary gland by mastitis pathogens, exists in three forms: clinical, subclinical, and chronic mastitis [6,7]. Among these forms, subclinical mastitis is more common and results in reduced milk production without observable clinical signs or milk abnormalities [8,9]. For this reason, it is difficult to diagnose and persists longer in the herd [10]. Subclinical mastitis (SCM) is the main form of this disease in dairy herds worldwide [11,12,13], and results in increased numbers of somatic cells in the produced milk and changes in its physical and chemical qualities [14].

The etiology of mastitis includes contagious microorganisms that survive and proliferate on the skin and teat wounds, as well as environmental microorganisms that are not retained on the teat [6,7,8]. More than 140 different pathogenic species have been reported [8]. Previously, studies had documented major pathogens of mastitis such as *Staphylococcus aureus*, *Streptococcus agalactiae*, and coliforms [15,16]. Current studies have reported a change in the pathogens from major pathogens to minor pathogens, such as coagulase-negative *Staphylococcus* and other bacilli [9,17,18]. These studies have shown that these minor pathogens may be playing a significant role in the pathogenesis of mastitis and vary from herd to herd [19,20].

The main treatment of mastitis is commonly administered by intramammary infusion or parenteral administration of antibiotics, such as streptomycin, ampicillin, cloxacillin, penicillin, and tetracycline [21]. The effective treatment of bovine mastitis depends on the antimicrobial susceptibility of the pathogens, the type of mastitis, the cattle breed, and the treatment regimen [22]. The emergence of drug resistance is a serious challenge for mastitis control, as resistance profiles are often herd specific [23]. Combining more than one synergistic antimicrobial agent may be more effective than using a single drug, and can achieve a high cure rate [4,20,24].

The prompt identification and understanding of the diversity of the pathogens associated with mastitis is essential for effective prevention and control [20]. However, the treatment is anticipated to become problematic in the near future owing to the rapid increase in antibiotic-resistant pathogens [20]. The transmission of antimicrobial resistant mastitis pathogens and foodborne pathogens to humans could occurred if unpasteurized milk is consumed [5,10,25]. The widespread use of antibiotics in the control of mastitis greatly increases the risk of installing and transmitting antibiotic resistance to consumers. Such a possibility is constantly in the attention of animal health and public health authorities, requiring a scientifically grounded redefinition of antibiotic therapies taking into account the intersection of animal welfare with social concerns [26,27].

The purpose of this study was to estimate the distribution of pathogens associated with clinical and subclinical mastitis and to determine their antimicrobial resistance patterns, in a random selection of dairy farms in the western part of Romania. To the authors’ knowledge, there is a lack of data on potential regional differences in the prevalence of different mastitis pathogens and their antimicrobial resistance in Romania.

## 2. Results

### 2.1. Microbiological Results

Cattle with clinical mastitis and subclinical mastitis were observed at all of the farms included in this study; samples from SCM cases represented 90% of the total number of samples that were collected. Clinical mastitis (CM) cases were obviously lower in all studied farms. The value of χ^2^ = 0.109, with a degree of freedom 3, confidence level of 95%, and *p* = 0.99, demonstrates that these two variables are significative associates.

The distribution of the most commonly isolated genera was: *Staphylococcus* spp., 43.1%; *Streptococcus* spp., 22.42%; *E. coli*, 13.79%; *Enterococcus* spp., 8.62%; *Corynebacterium* spp., 7.75%; and *Enterobacter* spp., 4.31% (Table 1).

The identified species of coagulase-positive staphylococci (CPS) represented 64% of all isolated *Staphylococcus* species as it follows:-*S. aureus,* twenty-four strains (48.0%);-*S. intermedius,* six strains (12%);-*S. hycus* subsp. *hycus,* two strains (4%);

Mahmoud et al. [28] isolated CPS at a percentage of 48.2%, since CNS (coagulase negative staphylococci) were isolated at a lower percentage, 8.9%.

The presence of a lower number of CPS strains other than *S. aureus* was observed in the literature by other researchers [29,30,31,32].

In our study, CNS—*S. hycus*, *S. chromogenes*, *S. xylosus*, and *S. capitis*—were identified with a lower proportion, and represented 36% (18/50) of the number of staphylococci isolated strains and 15.51% (18/116) of the total of isolated strains.

From the streptococci group, we isolated and identified thirteen strains of *Str. Agalactiae,* considered a species with major pathogenicity, six strains of *Str. uberis*, four strains of *Str. dysgalactiae* and ten strains of *Enterococcus* spp. The streptococci group represented 22.41% (26/116) and enterococci represented 8.62% (10/116) of all isolated strains.

*Corynebacterium bovis* and *Corynebacterium* spp. were isolated in a proportion of 7.75% of all isolates identified in this study. Most strains were isolated from SCM (7/9; 77.77%).

In this study, *E. coli* represented 13.79% (16/116), and *Enterobacter aerogenes* and *E. cloacae* represented 4.31% of all isolated strains.

### 2.2. Antimicrobial Susceptibility Assay

All 24 of the isolated CPS and CNS strains were found to be resistant to at least four antimicrobial agents. The isolated strains showed multiple resistance. Resistance was observed for ampicillin (17/24), polymyxin B (16/24), tetracycline (15/24), tylosin (14/24), amoxicillin-clavulanic acid, oxacillin, erythromycin, methicillin (12/24), and novobiocin (10/24). There was low resistance found for kanamycin, gentamicin, amoxicillin, and cephalothin. Other similar studies reported that cephalosporins could have an increased efficacy against *Staphylococcus* spp. isolates [33]. No resistance was found for rifampicin (Table 2). It is important to mention that all *S. aureus* strains which were tested with methicillin were resistant to this antimicrobial agent.

The phenomenon of multiple resistance to the tested antimicrobial substances was noted, in the present research, in which out of the 24 staphylococci strains tested for antibiotic resistance, 21 isolates (87.5%) presented multiple resistance: eight (33.3%) strains were resistant to four antimicrobials, four (16.66%) to five antimicrobials, five (20.83%) to six antimicrobials, two (8.33%) to seven antimicrobials, and two (8.33%) to eight antimicrobials. 

Behavior towards antimicrobials was tested for 14 of the strains of streptococci isolated from milk samples, the culture medium being supplemented with sheep blood, for safe and clear growth.

All 16 of the isolated streptococci strains were found to be resistant to almost all 14 antimicrobials tested (Table 3). *Str. agalactiae* showed the highest resistance, with resistance being observed towards novobiocin, tetracycline, kanamycin, bacitracin, and sulfamethoxazole-trimethoprim—total resistance was recorded for each (8/8), erythromycin and gentamycin (7/8), oxacillin (6/8), penicillin and lincomycin (5/8), amoxicillin-clavulanic acid and tylosin (4/8). *Str uberis* and *Str. dysgalactiae* showed the highest resistance towards novobiocin, tetracycline, kanamycin, bacitracin and sulfamethoxazole-trimethoprim.

*Enterococcus faecium* isolated strains showed resistance towards eight antimicrobials from the 14 tested.

None of the isolates were sensitive for all antimicrobials, an aspect also observed by others [34,35,36]. In the present research, all 14 strains tested showed the multiple resistance phenomenon, as follows: six strains (42.9%) to five antimicrobials, three strains (21.4%) to six antimicrobials, two strains (14.3%) to eight antimicrobials, two strains (14.3%) to nine antimicrobials and one strain (7.1%) to ten antimicrobials.

The phenomenon of multiple resistance to the tested antimicrobials was also noticed in the enterococci strains where four (66.6%) strains showed this phenomenon, as follows: one strain (16.6%) was resistant to four antimicrobials, one strain (16.6%) to six antimicrobials, one strain (16.6%) to seven antimicrobials, and one strain (16.6%) to all ten of the antimicrobials tested.

In *E. coli* strains (Table 4), resistance was observed for erythromycin and tetracycline (5/7) and ampicillin (4/7). The *E.* coli isolated strains were susceptible for enrofloxacin, gentamicin, and florfenicol, in which the highest sensitivity was noted (7/7). Other authors also [22,37] reported the high efficacy of gentamicin, enrofloxacin, and florfenicol.

All of the 39 studied isolates that belonged to several bacterial genera were resistant to at least three antimicrobial agents (Table 5). The phenomenon of resistance was manifested for several antibiotics, such as erythromycin, amoxicillin-clavulanic acid, ampicillin, tetracycline, and gentamicin, antibiotics which are used frequently in cow mastitis treatment in Romania, and not only.

## 3. Discussions

Our study agrees with the results of previous investigations [2,5,11,20,38,39]. Some factors must be considered when analyzing the prevalence of mastitis in dairy cows and the presence of multidrug resistance in bacteria isolated from mastitis. These factors include the health status of the animal, the season in which the samplings took place, the sampling method used, and the method used for the isolation and identification of the bacteria. 

*S. aureus* remain the main species identified in mastitis. According to different authors, *S. aureus* was isolated in up to 40% of mastitis cases in China [40] and other countries [19,23,32].

In our study, *S. aureus* were predominantly isolated from monomicrobial mastitis and *S. intermedius* was isolated only in monomicrobial mastitis. The occurrence of *S. intermedius* mastitis has been an increasing problem in Turkey. Additionally, the first report of a brain abscess in a human due to *S. intermedius* was mentioned in literature [41,42]. Given the zoonotic risk of *S. intermedius*, this staphylococcal species should be a concern as an etiological agent of subclinical mastitis in cows.

In SCM cases, *S. aures* was isolated in association with streptococci, coagulase-negative staphylococci, enterococci, and *Corynebacterium bovis*.

In a study conducted by Wald et al. [43], the CNS causing intramammary infections were mainly *S. xylosus* (24/40) and *S. chromogenes* (18/26) and were predominantly associated with a subclinical presentation. According to Pitkälä et al. [44], the percent of CNS isolated strains was 17% in Finland while Roberson et al. [45] observed a prevalence of 28% of mastitis produced by CNS in the USA. 

The prevalence of mastitis produced by *Str. agalactiae* mentioned in the literature is variable, between 6.8% and 14.4% [46,47]. The real incidence of mastitis produced by streptococci and enterococci is not well-known due to the phenotypical similitudes between streptococci and enterococci, similitudes which made the correct diagnosis of these mastitis difficult [20,30,48].

Gram-negative bacteria, mostly coliforms, including *E. coli*, *Klebsiella* spp., and *Enterobacter* spp., cause a high proportion of all clinical mastitis (CM) cases [30,48,49]. *E. coli* is the most common Gram-negative species causing CM in dairy cattle [15,17,50].

The low proportion of Gram-negative microorganisms isolated from the mastitis cases studied is indirectly proportional to the increased frequency of major pathogens (*S. aureus*, *Str. agalactiae*) involved in the etiology of mastitis in cows. This aspect observed in the present study is in concordance with the literature [17,46,47].

Antimicrobial resistance is a growing problem in cow mastitis. Antimicrobial resistance helps bacteria stay alive after treatment with antibiotics, and some of the mechanisms of resistance include the presence of antimicrobial resistance genes that can spread by horizontal transfer from bacteria to bacteria with mobile genetic elements such as plasmids, phages, and pathogenicity islands, or through random mutations when the bacteria are under stress [51,52]. In the cases of mastitis, the prevalence of antimicrobial resistant bacteria seems to be increasing at least for some antimicrobials. Studies reported that over 50% of isolates that cause mastitis were resistant to either beta-lactam drugs or penicillin [53].

Some authors noted that in the therapy of mastitis caused by staphylococci sensitive to penicillins, it is recommended to administer β-lactam antimicrobials (especially penicillin G), and as an alternative treatment, cloxacillin, macrolides, and lincosamides can be used. The same authors do not recommend the use of fourth generation cephalosporins as an alternative treatment, as they may generate strains resistant to broad-spectrum β-lactams [51,54,55].

All 12 *Staphylococcus* spp. isolates tested with methicillin, amoxicillin, and rifampicin demonstrate a total resistance to methicillin, which is a real concern for animal and human health. Staphylococcus methicillin resistance (MRSA) is one significant problem in society, being demonstrated that these strains can be transmitted to humans. Milk and milk producers may act as a reservoir of MRSA. An early determination of methicillin resistance is of crucial importance in the prognosis of *S. aureus* infections [51,52,56].

The *Staphylococcus* isolates were resistant to eight antimicrobials from the 14 tested. Literature mentioned the in vitro biofilm forming abilities of *S. aureus* and *S. epidermidis* isolates from bovine mastitis cases [57], and the possibility that this biofilm can influence their susceptibility to antimicrobial agents. Bacterial biofilms have been directly identified in bovine udders with mastitis. The continuous unsuccessful antibiotic treatment of potential biofilm mastitis infections can increase the risk of antibiotic resistance, which is one of the biggest threats to human and animal health [57,58]

Kaczorek et al. [55] reported that *Streptococcus* spp. are more resistant to gentamycin, kanamycin, and tetracycline, but highly susceptible to penicillin, enrofloxacin, and marbofloxacin. As in the case of staphylococci, also in the streptococci isolates, the phenomenon of multiple resistance to the tested antimicrobial substances was noted. In Finland, Pitkälä et al. [44] noted multiple resistance to 25.4% of 63 strains of *Enterococcus* spp. isolated from milk samples with the highest resistance to aminoglycosides.

Some of the Gram-negative environmental mastitis pathogens, such as *E. coli* and *Enterobacter* spp., are the greatest threats to human health due to the emergence of strains that are resistant to all or most available antimicrobials [59,60].

Quinn et al. [30] also recommends gentamicin in the therapy of mastitis caused by *E. coli*. In other experimental or clinical studies, the efficacy of enrofloxacin treatments has been demonstrated [38].

A clear difference between our results and those from other studies is the much higher proportion of environmental pathogens in many other countries. In studies in the UK, *Str. uberis* were the species most commonly isolated, followed by *Enterobacter*, while in Finland [20], *Str. agalactiae* was rarely isolated.

Monitoring antimicrobial resistance patterns of bacterial isolates from cases of mastitis is important for treatment decisions. The prudent use of antimicrobials in dairy farms reduces the emergence, persistence, and spread of antimicrobial-resistant bacterial strains and resistome from farms to animals, humans, and the environment.

## 4. Conclusions

The data from this study revealed that MRSA isolates are present among *S. aures* strains, which is a real concern for both animal and human health. *Staphylococcus* methicillin resistance is one significant problem in society, being demonstrated that these strains can be transmitted to human resistance. The need to implement the One Health concept is more urgent than ever, if we contemplate the interconnections between humans, animals, animal products, and the environment.

The multiple resistance phenomena observed in a high number of isolates require discern in the choice of mastitis treatment, considering both the health of the animal, the increase of productivity, but also the ease of transmitting bacteria from milk to humans. For this reason, the testing of antimicrobial sensitivities are strongly recommended.

We hypothesized that resistance pathotypes were more prevalent in the studied farms, not only due to exposure to a higher number of antibiotics, but also due to the greater frequency of isolation of bacterial strains with an increased resistance to antibiotics.

## 5. Materials and Methods

### 5.1. Study Farms

A total of 127 lactating cows in a random selection of 4 farms located in the western part of Romania were included in the present study (Table 6). The animals were a mixed breed (Holstein, Red Holstein, and Romanian Spotted Cattle). The cows differed in age, number of milking days, number of calves, and milk yield.

### 5.2. Samples Collection

A cross-sectional study was used to screen mastitis in lactating dairy cows from 15 May–30 September 2021. The sample size was determined according to Thrusfield [61] formula with 95% statistical confidence level. Accordingly, 127 lactating cows were included in the study. From these cows with/without characteristic clinical signs of mastitis, 180 milk samples were collected. All methods were conducted in accordance with relevant guidelines and regulations. As the samples were intended for diagnosis, the collection protocol was carried out with the consent of animal owners, according to the code of the Romanian Veterinary College (protocol numbers 34/1 December 2012) and the proper procedures of the Clinics in the Faculty of Veterinary Medicine Timisoara [62].

The main criteria for inclusion in this study were:Presence of signs of udder inflammation: redness, hotness, pain at local palpation.Obvious changes in milk (milk consistency, presence of blood, clots, and flakes).Generalized clinical symptoms: fever, loss of appetite, severe udder inflammation.

These criteria made statements on the clinical mastitis diagnosis. Subclinical mastitis cases were classified, as those cows were without clinical signs, but with a high somatic cell count determined using the California mastitis test (CMT). Reagent (CMT TEST, KEPRO, Deventer, The Netherlands) was mixed with an equal volume of milk (2 mL) in a four-well paddle for 10–15 s, and the result was recorded within 20 s. Results were interpreted using a scoring system ranging from 0 to 4: 0 for no reaction, 1 for trace, 2 for weakly positive (presence of a sediment), 3 for distinctly positive (sediment and a slight increase in consistency) and 4 for strongly positive (sample is totally coagulated) [63]. Milk samples were aseptically collected for bacteriological assays from clinical mastitis and subclinical mastitis cases, after the udder was washed clean and dried. After discarding the first 6 streams of milk, the teat ends were disinfected with swabs soaked in betadine, allowed to dry, and 10 mL of milk were collected in sterile tubes. Samples were refrigerated in iceboxes with cold packs and transported to the Laboratory of bacterial infections diagnosis, Faculty of Veterinary Medicine Timisoara, for processing. The samples were cultured immediately or stored in the refrigerator at 4 °C for a maximum of a day, awaiting culture.

### 5.3. Bacterial Isolation and Identification

Milk samples were analyzed microbiologically using standard laboratory methods [30]. A loopful (approximatively 0.01 mL) of milk was streaked onto the surface of blood agar (agar-based medium enriched with 5% sterile sheep blood) and MacConkey agar plates (Oxoid, Basingstoke, UK). The plates were incubated aerobically at 37 °C for 24–48 h. After which, the colony morphology was evaluated and recorded. Samples yielding more than one colony were grouped as mixed cultures. The distinct colonies were subcultured separately to obtain pure colonies by restreaking. Subcultures were made to obtain pure isolates for final identification. The pure isolates were identified using phenotyping tests, including Gram stain, oxidase, indole, and catalase tests. Identification of bacterial species was performed based on biochemical profiles using API system (API 20E, API Staph, API 20Strep, API NE, API Coryne, bioMerieux, Marcy l’Etoile, France) 

*Staphylococcus* spp. were identified based on the evidence of free coagulase and the presence of the clumping factor using rapid slide agglutination tests (Bactident Coagulase, Merk, Darmstadt, Germany) and Staphytec Plus (Oxoid, Basingstoke, UK). The colonies presenting phenotypical characteristics were transferred on Chapman agar (Oxoid, Basignstoke, UK) to identify pathogenic *Staphylococcus* strains (mannitol fermentation).

*Streptococcus* species were identified using the catalase test and growth characteristics on Edward’s media (Oxoid, Basingstoke, UK) and the within group differentiation was done using the CAMP test. Gram-negative bacteria were identified based on growth on MacConkey agar, motility, and indole and oxidase tests.

The isolates were sent to the other laboratory for identification with matrix-assisted laser desorption ionization time-of-flight (MALDI-TOF) mass spectrometry (MS).

For MALDI-TOF MS analysis, samples were prepared using a single colony of fresh overnight culture and were prepared according to standard protocol provided by the manufacturer (Bruker Daltonic, Bremen, Germany). A small amount of biological material from a single colony was smeared onto the spot of the MALDI target plate. Using the same instrument, a second spot was made on the target plate. The biological material was covered with 1 µL of matrix solution for 1 h and allowed to dry at room temperature. The target plate was then loaded on MALDI-TOF Biotyper and identified by comparing the mass spectral protein detection pattern with the reference patterns in database. A score of >0.2 was accepted as representing a reliable identification [34,64].

### 5.4. In Vitro Antibiotic Susceptibility Test

Antimicrobial susceptibility was evaluated using the disc diffusion (Kirby-Bauer) method. Briefly, representative strains of the isolated bacteria were spread on Mueller-Hinton agar plates (Difco), and their susceptibility to the following antibiotics (Oxoid, Basingstoke, UK) was tested. *Staphylococcus* spp. isolates were tested to: erythromycin (E; 30 μg/disc), polymyxin B (PB; 300 UI/disc), amoxicillin/clavulanic acid (2:1) (AMC; 30 µg/disc), ampicillin (AMP; 25 μg/disc), tylozin (TY; 30 µg/disc), oxacillin (OX; 1 µg/disc), cephalothin (CH; 30 µg/disc), novobiocin (NV; 5 µg/disc), gentamicin (GEN; 10 µg/disc), neomycin (NEO; 30 µg/disc), penicillin G (PEN; 10 units/disc), streptomycin (STR; 10 µg/disc), sulfamethoxazole/trimethoprim (SXT; 25 µg/disc), kanamycin (K; 30 µg/disc) and tetracycline (TET; 30 µg/disc). Twelve of these *Staphylococcus* isolates were also tested to methicillin (M; 10 µg/disc–HiMedia, India), amoxicillin (AC; 20 µg/disc, HiMedia, India), and rifampicin (R; 5 µg/disc, HiMedia, India). *Streptococcus* spp. isolates were tested to the same antimicrobials used for staphylococci isolates except polymyxin B. There were few antibiotics which were also added: lincomycin (L; 15 µg/disc), penicillin G (P; 10 IU/disc), bacitracin (B; 30 µg/disc), and sulfamethoxazole/trimethoprim (SXT; 25 µg/disc). Six strains of *Enterococcus* spp. were tested to erythromycin (E; 30 µg/disc), ampicillin (AMP; 25 µg/disc), tylozin (TY; 30 µg/disc), oxacillin (OX; 1 µg/disc), gentamicin (GEN; 10 µg/disc), sulfamethoxazole/trimethoprim (SXT; 25 µg/disc), kanamycin (K; 30 µg/disc), lincomycin (L; 15 µg/disc), penicillin G (P; 10 IU/disc), bacitracin (B; 30 µg/disc), and tetracycline (TET; 30 µg/disc).

*E. coli* isolates and *Enterobacter* spp. isolates were tested to: erythromycin (E; 30 µg/disc), florfenicol (FFC; 30 30 µg/disc), gentamicin (GEN; 10 µg/disc), cephalothin (CH; 30 µg/disc), tetracycline (TET; 30 µg/disc), ampicillin (AMP; 25 µg/disc), and enrofloxacin (ENR; 15 µg/disc).

The choice of antimicrobial substances for the antibiogram was mainly influenced by data from the literature [2,12,20,26] on the sensitivity of bacterial species and by the composition of various products intended for the treatment of mastitis, especially those administered intramammarily. In this way, it was also aimed to establish an effective therapy for the studied intramammary infections. 

Zones of inhibition (in mm) were measured after approximately 18 h of incubation at 37 °C, and the results were interpreted following Clinical and Laboratory Standards Institute [65] tables. The results are expressed in terms of susceptibility, intermediate and resistance, with the number of susceptible isolates out of the total number tested being given.

Quality control was performed following the guidelines specified by the CLSI (CLSI, 2008) using *Staphylococcus aureus* ATCC 23235 and *Escherichia coli* ATCC 25922. All susceptibility results obtained from quality control strains were within the quality control ranges.

## Figures and Tables

**Table 1 antibiotics-11-00057-t001:** Microorganisms isolated from mastitic milk and their distribution depending on the character of primary culture and mastitis evolution type.

Bacteria	Character of Primary Culture				Total of Isolates		Mastitis Type			
	Monomicrobial		Polymicrobial		No	%	CM		SCM	
	No Isolates	%	No Isolates	%			No	%	No	%
*Staphylococcus* spp.	29	25.02	21	18.10	50	43.10	13	11.20	37	31.90
*Streptococcus* spp.	19	16.37	7	6.03	26	22.41	11	9.48	15	12.94
*Escherichia coli*	9	7.75	7	6.03	16	13.80	10	8.62	6	5.17
*Corynebacterium* spp.	3	2.58	6	5.17	9	7.76	2	1.72	7	6.03
*Enterococcus* spp.	6	5.17	4	3.45	10	8.62	2	1.72	8	6.90
*Enterobacter* spp.	4	3.45	1	0.86	5	4.31	-	-	5	4.31
Total	70	60.34	46	39.66	116	100	38	32.74	78	67.25

**Table 2 antibiotics-11-00057-t002:** Antimicrobial drug resistance profile of the isolated staphylococci strains.

No	Antimicrobial	CPS Isolates			CNS Isolates			
		*S. aureus* (*n* = 12)	*S. intermedius* (*n* = 3)	*S. hycus* subsp. *Hycus* (*n* = 1)	*S. chromogenes* (*n* = 2)	*S. xylosus* (*n* = 2)	*S. hycus* (*n* = 2)	*S. capitis* (*n* = 2)
1	Erythromycin	9	1	1	0	1	0	0
2	Polymyxin B	8	2	0	1	2	1	2
3	Amoxicillin-clavulanic acid	3	3	1	2	1	2	0
4	Gentamicin	0	1	0	1	1	0	1
5	Tylozin	6	2	0	2	2	1	1
6	Oxacillin	6	3	1	1	0	1	0
7	Cephalothin	0	0	0	1	0	0	0
8	Novobiocin	5	2	0	1	1	1	0
9	Ampicillin	10	2	1	1	1	1	1
10	Tetracyclin	8	3	0	1	2	1	0
11	Kanamycin	2	1	0	1	2	1	0
12	Methicillin	12	0	0	0	0	0	0
13	Amoxicillin	1	0	0	0	0	0	0
14	Rifampicin	0	0	0	0	0	0	0

**Table 3 antibiotics-11-00057-t003:** Antimicrobial drug resistance profile of the isolated streptococci and enterococci strains.

No	Antimicrobial	*Streptococcus* spp.			*Enterococcus* spp.
		*Str. agalactiae* (*n* = 8)	*Str. Uberis* (*n* = 4)	*Str. Dysgalactiae* (*n* = 4)	*E. faecium* (*n* = 6)
1	Erythromycin	7	1	0	4
2	Penicillin	5	1	1	3
3	Amoxicillin-clavulanic acid	4	3	1	5
4	Gentamicin	7	3	4	5
5	Tylosin	4	2	3	4
6	Oxacillin	6	3	3	1
7	Cephalothin	0	0	0	1
8	Novobiocin	8	4	4	2
9	Ampicillin	1	1	0	1
10	Tetracycline	8	4	4	4
11	Kanamycin	8	4	4	5
12	Lincomycin	5	3	3	1
13	Bacitracin	8	4	4	1
14	Sulfamethoxazole/trimethorprim	8	4	4	6

**Table 4 antibiotics-11-00057-t004:** Antimicrobial drug resistance profile of the isolated *E. coli* strains.

No	Antimicrobial	*E. coli* (*n* = 7)
1	Erythromycin	5
2	Florfenicol	0
3	Gentamicin	0
4	Cephalothin	2
5	Ampicillin	4
6	Tetracycline	5
7	Enrofloxacin	0

**Table 5 antibiotics-11-00057-t005:** Patterns of resistance identified in *Staphylococcus* spp. (*n* = 21), *Streptococcus* spp. (*n* = 14) and *Enterococcus* spp. (*n* = 4) isolates.

Bacteria	No of Isolates	Resistance Profile
*Staphylococcu* spp.	8	E;AMC;TY;AMP
	4	E;AMC;AMP;TC;M
	5	M;TC;AMP;NV;E
	2	AMC;PB;OX;NV;AMP;TET;M
	2	PB;GEN;TYL;NV;AMP;TET;M;OX
*Streptococcus* spp.	6	GEN;OX;TET;K;STX
	3	E;AMP;TET;K;B;STX
	2	E;B;L;K;TET;NV;OX;AMC
	2	STX;B;L;K;TET;NV;OX;TYL;AMC
	1	E;AMC;GEN;OX;NV;AMP;TET;K;B;STX
*Enterococcus* spp.	1	E;GEN;K;STX;
	1	STX;K;TET;E;GEN;TY
	1	E;GEN;TY;TET;K;STX
	1	E;GEN;TY;OX;AMP;TET;K;L;B;STX

Legend: Erythromycin (E), Polymyxin B (PB), Amoxicillin/clavulanic acid (AMC), Ampicillin (AMP), Tylozin (TY), Oxacillin (OX), Cephalothin (CH), Novobiocin (NV), Gentamicin (GEN), Neomycin (NEO), Penicillin G (PEN), Streptomycin (STR), Sulfamethoxazole/trimethoprim (SXT), Kanamycin (K), Tetracycline (TET), Methicillin (M), Amoxicillin (AC), Lincomycin (L).

**Table 6 antibiotics-11-00057-t006:** Distribution of farms and animals sampled in the study.

Farm	Location	Breed	Herd Size/Lactating Cows	No. of Samples		
				Total	From CM Cases (%)	From SCM Cases (%)
1	Arad	Holstein	160/50	102	9 (5.00)	93 (51.67)
2	Timis 1	Holestein and Red Holstein	75/33	31	5 (2.78)	26 (14.44)
3	Timis 2	Holestein	60/24	27	3 (1.67)	24 (13.33)
4	Bihor	Romanian Spotted Cattle	40/20	20	1 (0.56)	19 (10.56)
Total			335/127	180	18 (10.00)	162 (90.00)

## Data Availability

The datasets generated and analyzed during the current study are included within the article.

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
