# Peer review of "Etiology of Mastitis and Antimicrobial Resistance in Dairy Cattle Farms in the Western Part of Romania"

_antibiotics, 2022, doi:10.3390/antibiotics11010057_

Round 1

Reviewer 1 Report

The manuscript is a descriptive study about the occurrence of mastitis in 127 lactating cows from four farms from the western part of Romania. The study was based on the selection (no criteria described) of 4 farms and 127 cows for collection of milk samples for microbiological culture and Maldi-tof, and results presented as the frequency of isolation as isolates were submitted to antimicrobial susceptibility testing. In general terms, the study is deficient in terms of originality and the significance of the content, considering the experimental design used (convenience sampling of farms and cows) and because the etiology of bovine mastitis is extensively well studied and reported in previous studies. Additionally, it was not clear which was the research question of the study and which wasthe contribution the results would bring this field. Even though, bacterial isolates were submitted to Maldi-tof identification no clear and detailed description of isolation at the species level were provided (Table 1) and antimicrobial susceptibility testing was performed in isolates only described at the genus level, which is a main limitation for the interpretation of the results.

Reviewer 2 Report

The introduction focuses on very basic issues which are well-known. It should be completely rewritten, focusing on resistance rather than on mastitis.

Also, the hypothesis of the authors should be clearly described.

The objectives of the study are missing.

The study is a rudimentary work that does not guarantee publication in a high-class journal. The authors do not really offer any novel findings and really do not contribute to the advancement of knowledge. The study is a simple, small-scale, descriptive work of the type that was performed in the 1980’s and 1990’s.

Moreover, there many wrong expressions in scientific terminology and in English language (e.g., monomicrobial / polymicrobial).

However, I can understand the requirements for publishing as a means for advancing academic career and I fully support the authors in their endeavours. Therefore, I would agree to a revised, significantly shorter version of this work, about 2-3 pages maximum, that could be submitted as a letter to the editor, just presenting the salient features of the study.

Reviewer 3 Report

Major concerns

1- There are no line numbers to provide comments point by point

2- The introduction is long and not organized

3- you need to test the resistant isolates for the resistant gene of each antibiotic and provide correlations between genotypic and phenotypic analysis as provided in other previous studies (https://www.mdpi.com/2079-6382/10/12/1450

OR https://www.nature.com/articles/s41598-018-23962-7)

4- You need to separate the results and the discussion sections.

Other comments

1- The authors did not provide the aim of the study. Please add it to the end of the introduction

2- what is the sampling type. is it a cross-sectional study or what?

3- Provide reference for your scoring scale

4-  You mentioned that you did a lot of identification tests for the samples without providing details about the procedures of these tests.

5-Provide the source of antimicrobial discs

6- Statistical analysis needs to be provided in a separate section

7- The name of all the bacteria needs to be italic throughout the manuscript

Round 2

Reviewer 2 Report

The manuscript is of low value and does not really offer any novel contribution in the international literature. The authors did not improve substantially the manuscript after revision, hence it is borderline for rejection.

However, at this stage, it is not fair for the authors to receive a rejection opinion - this should have been done at an earlier stage of the evaluation.
Subject to improving language in the entire manuscript, then it can be accepted.

Reviewer 3 Report

Thank you for addressing the reviewer comments